# Dissipative cooling induced by pulse perturbations

A. Nava[1*], M. Fabrizio[2]

**1** Dipartimento di Fisica, Università della Calabria, Arcavacata di Rende I-87036, Cosenza, Italy
INFN - Gruppo collegato di Cosenza, Arcavacata di Rende I-87036, Cosenza, Italy
**2** International School for Advanced Studies (SISSA), Via Bonomea 265, I-34136 Trieste, Italy
* andrea.nava@fis.unical.it

April 26, 2021

## Abstract

We investigate the dynamics brought on by an impulse perturbation in two infinite-range quantum Ising models coupled to each other and to a dissipative bath. We show that, if dissipation is faster the higher the excitation energy, the pulse perturbation cools down the low-energy sector of the system, at the expense of the high-energy one, eventually stabilising a transient symmetry-broken state at temperatures higher than the equilibrium critical one. Such non-thermal quasi-steady state may survive for quite a long time after the pulse, if the latter is properly tailored.

## 1 Introduction

Shooting ultrashort laser pulses has emerged in the last decades as a new fast-driving tool for phase transformations, with great potentials especially for strongly correlated materials [1–3, 3–9], whose phase diagrams include close-by insulating, conducting and even

superconducting phases. Moreover, firing a laser pulse at a strongly correlated material not always boils down to a fast rise of the internal temperature, as one would reasonably expect. For instance, it sometimes allows uncovering hidden states inaccessible at thermal equilibrium [10, 11]. However, until now the most remarkable failure of the naïve correspondence between light firing and thermal heating is the evidence of superconducting-like behaviour at nominal temperatures far higher than the critical one in the molecular conductors $K_3C_{60}$ and $\kappa$-(BEDT-TTF)$_2$Cu[N(CN)$_2$]Br irradiated by laser pulses [12–14]. Such non-thermal state is transient, but may become rather long-lived by properly tailoring the laser pulse [14]. Even though this phenomenon may have explanations that concern material-specific mechanisms [12, 13], still it is legitimate to address the general question whether a laser pulse could ever cool down a solid state material. Spontaneous anti-Stokes emission of photons with higher energies than those absorbed from the incident light is a known laser cooling mechanism for semiconductors [15–19]. However, the same mechanism would not work in metals at low temperatures, e.g., in the above mentioned molecular conductors, where most of the entropy is carried by the electrons, and not by the phonons as in semiconductors.

In an attempt to explain the photoinduced superconductivity in $K_3C_{60}$ [12], a different laser cooling mechanism was proposed in Ref. [20], which is essentially based on the existence of a high energy localised mode that, when the laser is on, is able to fast soak up entropy from the thermal bath of low-energy particle-hole excitations, while, after the end of the laser pulse, it release back that absorbed entropy very slowly. It follows that the population of particle-hole excitations gets reduced, as if its internal temperature were lower, for a transient time after the pulse that is longer the smaller the non-radiative decay rate of the high energy mode. This idea was later tested [21] with success in a fully-connected toy model subject to a time dependent perturbation of finite duration, mimicking a 'laser pulse'. This model is trivially solvable since infinite connectivity implies that mean-field theory is exact in the thermodynamic limit. However, this feature, though providing the exact out-of-equilibrium dynamics, yet prevents full thermalisation, since it lacks internal dissipation. Therefore, despite the model does realise the laser cooling mechanism proposed in [20], one cannot exclude that dissipation could wash it out.

The aim of the present work is just to assess the role of dissipation in that same model. Since we want to maintain its mean-field character, we still assume full connectivity. Therefore dissipation cannot arise from the internal degrees of freedom, but it is simple included in the dynamics via a Lindblad equation. When all system excitations dissipate equally fast, we find just a quick relaxation to thermal equilibrium. On the contrary, if excitations dissipate faster the higher the energy, which is the most common physical situation, we do observe a transient regime where the low energy sector of the model effectively cools down. Moreover, if we increase the 'laser pulse' duration keeping constant the total energy supplied to the system, following the experiment in [14], we also find the transient state to last longer, not in disagreement with that experiment.

The paper is organized as follows. In Section 2 we introduce the quantum Ising model we shall investigate, an effective two-spin spin boson model [22] and its behavior in absence of dissipation. In Section 3 we briefly discuss the Lindblad master equation to describe the relaxation dynamic and the physical results obtained for different bath model. Finally, Sec. 4 is devoted to concluding remarks.

## 2 The model Hamiltonian

We consider the Hamiltonian of two coupled fully-connected quantum Ising models

$$\hat{H} = \sum_{n=1}^{2} \hat{H}_n - \lambda \sum_{j=1}^{N} \sigma_{1,j}^x \sigma_{2,j}^x, \tag{1}$$

where

$$\hat{H}_n = -\frac{J}{2N} \sum_{i,j=1}^{N} \sigma_{n,i}^x \sigma_{n,j}^x - h_n \sum_{i=1}^{N} \sigma_{n,i}^z, \tag{2}$$

and $\sigma_{n,i}^\alpha$, $\alpha = x, y, z$, are Pauli matrices on site $i = 1, \dots, N$ of the submodel $n = 1, 2$. All parameters, $J$, $h_1$, $h_2$ and $\lambda$ are assumed positive. Hereafter we take $J = 1$ as energy unit. Because of full connectivity, and for any $i \neq j$,

$$\left\langle \sigma_{n,i}^\alpha \sigma_{m,j}^\beta \right\rangle - \left\langle \sigma_{n,i}^\alpha \right\rangle \left\langle \sigma_{n,j}^\beta \right\rangle \propto \frac{1}{N}, \tag{3}$$

which actually implies that the mean-field approximation becomes exact in the thermodynamic limit $N \to \infty$, or, equivalently, that the full density matrix $\hat{\rho}$ factorises in that limit into the product of single-site density matrices:

$$\hat{\rho} \xrightarrow[N \to \infty]{} \prod_{i=1}^{N} \hat{\rho}_i, \tag{4}$$

where $\hat{\rho}_i$ are positive definite $4 \times 4$ matrices with unit trace. The property (4) allows exactly solving with relative ease the model Hamiltonian (1).

### 2.1 Equilibrium phase diagram

At equilibrium, one can exploit the variational principle for the free energy at temperature $T$,

$$F(T) = \min_{\hat{\rho}} \left[ \operatorname{Tr}\left( \hat{\rho} \, \hat{H} \right) + T \operatorname{Tr}\left( \hat{\rho} \ln \hat{\rho} \right) \right], \tag{5}$$

to find, through Eq. (4), the single-site density matrices $\hat{\rho}_i(T)$ that minimise the r.h.s. of Eq. (5), and which actually solve the self-consistency set of equations

$$\hat{\rho}_i(T) = \frac{\mathrm{e}^{-\beta \hat{H}_i(T)}}{\operatorname{Tr}\left( \mathrm{e}^{-\beta \hat{H}_i(T)} \right)},$$

$$\hat{H}_i(T) = -\sum_{n=1}^{2} \left[ J_{n,i}(T) \, \sigma_{n,i}^x + h_n \, \sigma_{n,i}^z \right] - \lambda \, \sigma_{1,i}^x \sigma_{2,i}^x \tag{6}$$

$$\equiv \sum_{n=1}^{2} \hat{H}_{n,i} - \lambda \, \sigma_{1,i}^x \sigma_{2,i}^x,$$

where

$$J_{n,i}(T) = \lim_{N \to \infty} \frac{1}{N} \sum_{j=1}^{N} \operatorname{Tr}\left( \hat{\rho}_j(T) \, \sigma_{n,j}^x \right) \equiv J_n(T). \tag{7}$$

Since $J_{n,i}(T) = J_n(T)$ is the same for all sites, so $\hat{\rho}_i(T)$ and $\hat{H}_i(T)$ in (6) are. Therefore, we can also write

$$J_{n,i}(T) = \operatorname{Tr}\left( \hat{\rho}_i(T) \, \sigma_{n,i}^x \right) \equiv m_{x,n}(T), \quad \forall \, i, \tag{8}$$

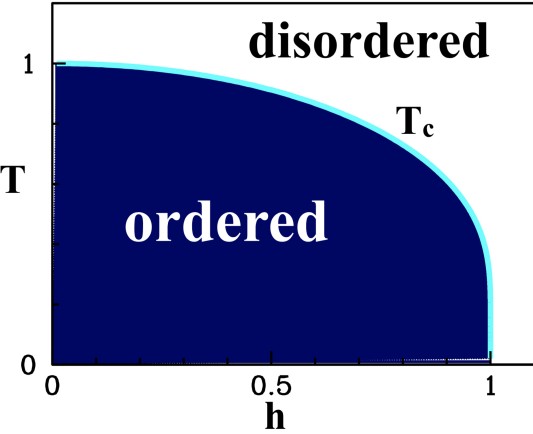

Figure 1: Phase diagram of the fully connected quantum Ising model (2) as a function of the transverse field $h$. In the blue coloured region, below the critical temperature, the system is in the ordered phase with spontaneously broken $Z_2$ symmetry.

which, together with Eq. (6), give an equivalent representation of the self-consistency equations.

At $\lambda = 0$ in Eq. (1), each Ising model (2) has the phase diagram shown in Fig. 1. For $h_n \leq 1$ and temperature

$$T \leq T_c(h_n) = \frac{2h_n}{\ln \dfrac{1+h_n}{1-h_n}} \; , \tag{9}$$

the expectation value $m_{x,n}(T)$ of $\sigma_{n,i}^x$, see Eq. (8), is finite, thus the model $n$ is in an ordered phase that spontaneously breaks the $Z_2$ symmetry $\sigma_{n,i}^x \to -\sigma_{n,i}^x, \; \forall i$. Above $T_c(h_n)$ or if $h_n > 1$, the $Z_2$ symmetry is restored, and the order parameter $m_{x,n}(T)$ vanishes identically. The mean-field Hamiltonian $\hat{H}_{n,i}$ of each subsystem $n = 1, 2$, see Eq. (6), has two eigenstates separated by an energy

$$E_n(T) = 2\sqrt{m_{x,n}(T)^2 + h_n^2} \; , \tag{10}$$

which corresponds to a dispersionless optical excitation branch of the Hamiltonian $\hat{H}_n$ in Eq. (2). For $h_n \leq 1$, $E_n(0) = 2$ at $T = 0$, and diminishes with $T$ until, at $T = T_c(h_n)$ and above, $E_n(T) = 2\,h_n$.

Throughout this work we assume

$$\lambda \ll h_1 < 1 \ll h_2 \,, \tag{11}$$

and, specifically,

$$\lambda = 10^{-2}, \quad h_1 = 0.5, \quad h_2 = 10. \tag{12}$$

In this case, for $T \leq T_c \simeq T_c(h_1)$, model 1 acquires a finite order parameter $m_{x,1}(T)$, which, in turn, drives a finite

$$m_{x,2}(T) \simeq \frac{\lambda}{h_2}\, m_{x,1}(T) \ll m_{x,1}(T) \,. \tag{13}$$

It follows that the Hamiltonian (1) has, at leading order in $\lambda$, two dispersionless excitation branches with energies

$$E_1(T) \simeq 2\sqrt{m_{x,1}(T)^2 + h_1^2} \; , \tag{14}$$
$$E_2(T) \simeq 2\,h_2 \gg \Delta E_1(T) \,.$$

In other words, we can write

$$\hat{H} \simeq \sum_{i=1}^{N} \sum_{n=1}^{2} E_n(T)\, b_{n,i}^{\dagger}\, b_{n,i}\,, \tag{15}$$

with $b_{n,i}$ hard core bosons. The local Hilbert space at site $i$ thus comprises four eigenstates

$$| k;i \rangle \equiv \left(b_{1,i}^{\dagger}\right)^{n_1(k)} \left(b_{2,i}^{\dagger}\right)^{n_2(k)} |0\rangle\,, \quad k = 0,\ldots,3\,, \tag{16}$$

with energy

$$E(k) = n_1(k)\, E_1(T) + n_2(k)\, E_2(T)\,, \tag{17}$$

where $n_2(k) = \lfloor k/2 \rfloor$ is the integer part of $k/2$, and $n_1(k) = k - 2n_2(k)$. The states $|0;i\rangle$ and $|1;i\rangle$ define a low energy subspace well separated from the high energy one, which includes states $|2;i\rangle$ and $|3;i\rangle$. It follows that there exists a wide temperature interval, $T_c \lesssim T \ll E_2(T) = 2h_2$, where the low energy sector is entropy rich, contrary to the high energy one, which practically bears no entropy.

## 2.2 Cooling strategy

Based on the last observation, Ref. [21] devised a strategy to exploit the high energy sector as an entropy sink able to cool down the low energy one, which we briefly sketch in this section.

We assume that the system is prepared in the equilibrium state corresponding to a temperature $T_c \lesssim T \ll 2h_2$. Its initial density matrix $\hat{\rho}(0)$ is therefore defined through Eq. (4), with $\hat{\rho}_i$ the solution of the self consistency equations (6) and (8). At $t = 0$ the following perturbation is turned on

$$\hat{V}(t) = -E(t)\, \cos\omega t \sum_{i=1}^{N} \sigma_{1,i}^{x}\, \sigma_{2,i}^{x}\,, \tag{18}$$

which mimics a laser pulse, whose envelope we hereafter parametrise as

$$E(t) = \left(\frac{t}{\tau}\right)^2 \exp\left[1 - \frac{1}{E_0} \left(\frac{t}{\tau}\right)^2\right]\,, \tag{19}$$

thus corresponding to a pulse of duration $\tau$ and peak amplitude $E_0$ achieved for $t_{max} = \tau\sqrt{E_0}$. We further assume the 'laser' frequency $\omega = E_2(T) - E_1(T)$, see Eq. (14), in resonance with the excitation process $b_{2,i}^{\dagger}\, b_{1,i} = |2;i\rangle\langle 1;i|$, and the hermitean conjugate de-excitation one. The rationale behind this choice is the following. If $p_k(T) = \langle\, |k;i\rangle\langle k;i|\,\rangle$ is the initial occupation probability, i.e. the equilibrium one at temperature $T$, of the state $|k;i\rangle$, then, for $T_c \lesssim T \ll 2h_2$, the high-energy sector starts almost unoccupied, $p_2(T) \simeq p_3(T) \simeq 0$, while

$$1 \geq \frac{p_1(T)}{p_0(T)} = \frac{1 - \tanh\beta\,h_1}{1 + \tanh\beta\,h_1} \gtrsim \frac{p_1(T_c)}{p_0(T_c)} = h_1\,. \tag{20}$$

The effect of the 'laser pulse' (18) is primarily to increase the formerly negligible $p_2$ by reducing $p_1$, eventually making the ratio $p_1/p_0$ drop beneath the threshold value $h_1$, below which $Z_2$ symmetry breaking, which is mostly a matter of the lower energy sector, spontaneously sets in. In other words, the system starts in the disordered phase above $T_c$ and, after the 'laser pulse', it may end up into the ordered one, as if it were cooler than it was initially. In reality, the energy lost by the low-energy sector plus that soaked up from the

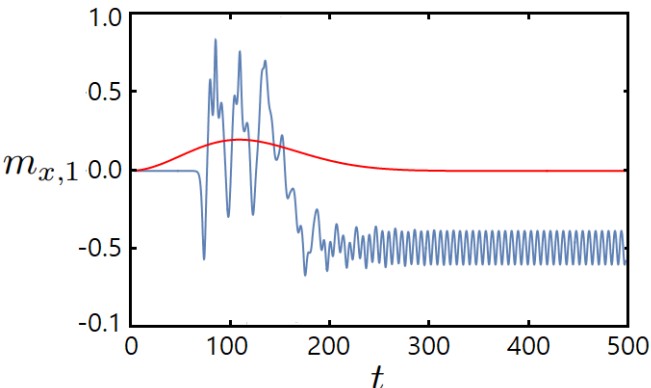

Figure 2: Time evolution of the order parameter of the low-energy sector (blue curve). The red curve corresponds to the envelope $E(t)$ of the perturbation, see Eq. (19), with $E_0 = 0.20$ and $\tau = 250$.

'laser pulse' is being temporarily stored in the high-energy sector, from which it later flows back and heats up the whole system, though gradually since $\lambda$ is tiny. Therefore, the low energy sector is transiently deprived of entropy by the 'laser pulse', for a time longer the small $\lambda$ is.

This scenario, put forth in Ref. [21], can be readily shown to occur in the model Hamiltonian (1) in presence of the perturbation (18). Indeed, since the full time-dependent Hamiltonian $\hat{H}(t) = \hat{H} + \hat{V}(t)$ does not spoil Eq. (3), the time-dependent density matrix $\hat{\rho}(t)$ can be still written as in Eq. (4) with $\hat{\rho}_i \to \hat{\rho}_i(t)$, where the latter evolves according to the first order non-linear differential equation,

$$\frac{\partial \hat{\rho}_i(t)}{\partial t} = -i \left[ \hat{H}_i(t) \,, \hat{\rho}_i(t) \right], \tag{21}$$

where, similarly to Eqs. (6) and (8),

$$\hat{H}_i(t) = -\sum_{n=1}^{2} \left[ J_{n,i}(t)\, \sigma_{n,i}^x + h_n\, \sigma_{n,i}^z \right] \\ - \left( \lambda + E(t)\, \cos \omega t \right) \sigma_{1,i}^x\, \sigma_{2,i}^x, \tag{22}$$

and the non-linearity arises because

$$J_{n,i}(t) = \text{Tr}\!\left( \hat{\rho}_i(t)\, \sigma_{n,i}^x \right), \tag{23}$$

is function of $\hat{\rho}_i(t)$. Eq. (21) must be solved with initial condition $\hat{\rho}_i(t=0) = \hat{\rho}_i(T)$, which, being actually the same for all sites $i$, implies that also $\hat{\rho}_i(t>0)$ is site-independent. Therefore one just needs to solve (21) for a single-site.
In Fig. 2 we show the time evolution of the low-energy sector order parameter $m_{x,1}(t)$ starting from the disordered equilibrium phase at $T = 1.5\,T_c$, and using, besides the Hamiltonian parameters in Eq. (12), a 'laser pulse' of duration $\tau = 250$ and amplitude $E_0 = 0.20$, see Eq. (19). We note that initially $m_{x,1} = 0$, since the system is disordered. However, after the 'laser pulse' the low energy sector ends up trapped into one of the two $Z_2$-equivalent symmetry variant phases, in the figure that with $m_{x,1}$ negative.

# 3  Dissipative dynamics

We already mentioned that the integrability of the Hamiltonian (1) has as counterpart the lack of any internal dissipation, as evident by the undamped oscillations in Fig. 2. This evidently raises doubts about the general validity of the results in the previous section. We could add internal dissipation giving up the possibility of exactly solving the model, e.g., by defining it on a lattice, and making the exchange $J$ in Eq. (1) decaying with the lattice distance between two spins [23]. However, even in such case the model would remain simply a toy one, unable to describe any real solid-state material. For this reason, we prefer to maintain full connectivity, and introduce local dissipation via the Lindblad formalism. Similarly, we do not pretend to derive the Lindblad equations from any Hamiltonian of the system plus a bath, upon integrating out the latter. Instead, we here consider the most general Lindblad equation compatible with the mean-field character of the Hamiltonian (1), and able to drive the system towards thermal equilibrium; specifically, compare with Eq. (21),

$$
\begin{aligned}
\frac{\partial \hat{\rho}_i(t)}{\partial t} = &-i \left[ \hat{H}_i(t) , \hat{\rho}_i(t) \right] \\
&+ \sum_{n<m} \Bigg[ \gamma_{n \leftarrow m}(t) \left( 2 \hat{L}_{n \leftarrow m}(t) \hat{\rho}_i(t) \hat{L}_{n \rightarrow m}(t) - \left\{ \hat{L}_{n \rightarrow m}(t) \hat{L}_{n \leftarrow m}(t) , \hat{\rho}_i(t) \right\} \right) \\
&+ \gamma_{n \rightarrow m}(t) \left( 2 \hat{L}_{n \rightarrow m}(t) \hat{\rho}_i(t) \hat{L}_{n \leftarrow m}(t) - \left\{ \hat{L}_{n \leftarrow m}(t) \hat{L}_{n \rightarrow m}(t) , \hat{\rho}_i(t) \right\} \right) \Bigg],
\end{aligned}
\tag{24}
$$

where $\hat{H}_i(t)$ is still defined through Eqs. (22) and (23), while the Lindblad downward jump operators are

$$
\hat{L}_{n \leftarrow m}(t) \equiv |n; i, t\rangle \langle m; i, t|, \quad n < m,
\tag{25}
$$

where $|n; i, t\rangle$, $n = 0, \ldots, 3$, is the instantaneous eigenstate of $\hat{H}_i(t)$ with eigenvalue $E_n(t)$, such that $E_0(t) \leq E_1(t) \leq E_2(t) \leq E_3(t)$. Assuming $n < m$, we distinguish between

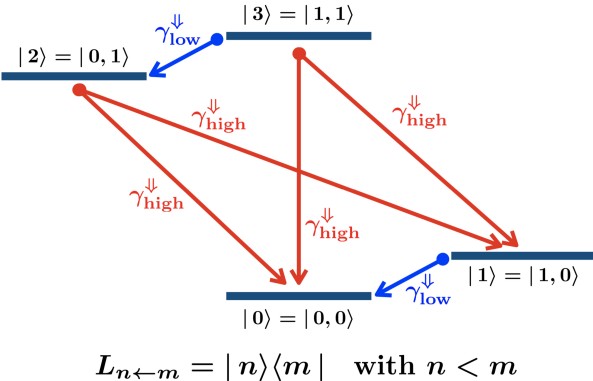

$$L_{n \leftarrow m} = |n\rangle\langle m| \quad \text{with } n < m$$

Figure 3:  Downward jump operators, $L_{n \leftarrow m} = |n\rangle\langle m|$ with $n < m$, which, together with their hermitean conjugates, $L_{n \leftarrow m}^\dagger \equiv L_{n \rightarrow m}$, the upward jump operators, define the dissipative Lindblad dynamics. We distinguish high energy jump operators, in red, from low-energy ones, in blue. The former have coupling strength $\gamma_{\text{high}}^{\Downarrow}$, while the latter $\gamma_{\text{low}}^{\Downarrow}$.

downward jump operators, $\hat{L}_{n \leftarrow m}(t)$, see Fig. 3, which correspond to de-excitations from a state to a lower energy one, and the reverse upward ones, $\hat{L}_{n \leftarrow m}^\dagger(t) \equiv L_{n \rightarrow m}(t)$. Detailed balance, which ensures that the Boltzmann distribution is the stationary solution of

Eq. (24), implies that

$$\gamma_{n\to m}(t) = \mathrm{e}^{-\beta\left(E_m(t)-E_n(t)\right)} \gamma_{n\leftarrow m}(t)\,, \tag{26}$$

where, since $n < m$, then $E_m(t) - E_n(t) > 0$, and therefore $\gamma_{n\to m}(t) < \gamma_{n\leftarrow m}(t)$. It follows that the Lindblad dynamics can be parametrised only through the six coupling strengths of the downward jump operators, $\gamma_{n\leftarrow m}(t)$ with $n < m$. In order to simplify the analysis, we assume that $\gamma_{n\leftarrow m}(t) = \gamma_{\mathrm{high}}^{\Downarrow}$ for all the high-energy de-excitation processes $(n,m) = (0,2),(0,3),(1,2),(1,3)$, red arrows in Fig. 3, distinct from $\gamma_{n\leftarrow m}(t) = \gamma_{\mathrm{low}}^{\Downarrow}$ for the low-energy ones $(n,m) = (0,1),(2,3)$, blue arrows in Fig. 3.

Since we expect that high-energy excitations dissipate faster than low-energy ones, we assume

$$\frac{\gamma_{\mathrm{high}}^{\Downarrow}}{\gamma_{\mathrm{low}}^{\Downarrow}} = r \geq 1\,. \tag{27}$$

Moreover, being the Lindblad equation (24) valid when the coupling to the dissipative bath is weak, we further take $\gamma_{\mathrm{high}}^{\Downarrow} = 0.05$ small, so that all other coupling strengths, $\gamma_{\mathrm{low}}^{\Downarrow}$ and the upward ones, see Eq. (26), are even smaller.

## 3.1 Time evolution upon changing bath and pulse parameters

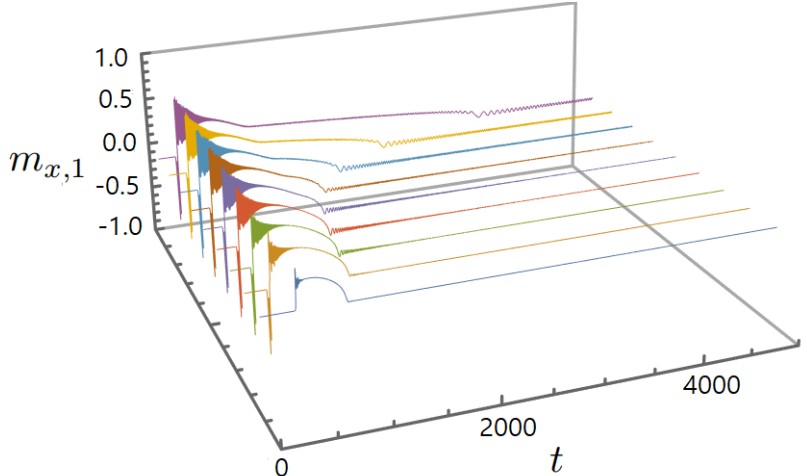

Figure 4: Time evolution of the order parameter $m_{x,1}$, compare with Fig. 2, for $r$ in Eq. (27) sets to $r = 1$ (blue curve), $r = 5$ (orange curve), $r = 10$ (green curve), $r = 20$ (red curve), $r = 40$ (blue curve), $r = 80$ (brown curve), $r = 160$ (light-blue curve), $r = 320$ (yellow curve) and, finally, $r = 640$ (purple curve). The pulse parameters are $E_0 = 0.2$ and $\tau = 1000$, see Eq. (19).

In Fig. 4 we show the results of the numerical integration of Eq. (24) at temperature $T = 1.5\,T_c$, and for increasing $r$, see Eq. (27), from $r = 1$, blue line, to $r = 640$, purple line. The laser pulse parameters, see Eq. (19), are $E_0 = 0.2$ and $\tau = 1000$. We note that when low-energy excitations dissipate as fast as high-energy ones, blue line at $r = 1$, the system quickly relax to thermal equilibrium, $m_{x,1} = 0$, without showing any transient cooling. The latter appears only upon increasing $r$, and lasts longer the larger $r$ is.

To quantify how long the system remains trapped into a non-thermal symmetry-broken state, we define a 'critical time' $t_{c,m}$ as the interval between the peak of the pulse, i.e.,

$t_{max} = \tau\sqrt{E_0}$, and the the time at which the magnetisation reaches its thermal value $m_{x,1} = 0$. Fig. 5 shows $t_{c,m}$ as a function of $r$, at fixed $E_0 = 0.14$ but different pulse

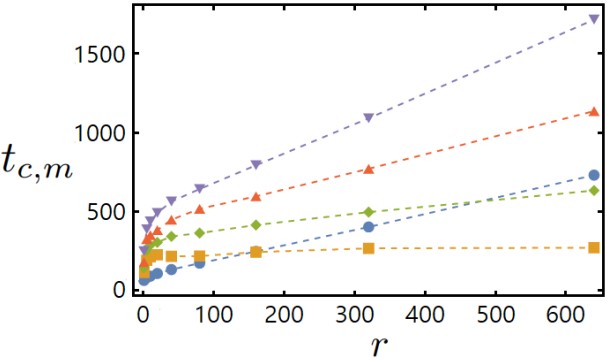

Figure 5: Critical time $t_{c,m}$ for $E_0 = 0.14$ and: $\tau = 250$ (blue circles), $\tau = 500$ (orange squares), $\tau = 650$ (green diamonds), $\tau = 800$ (red up triangles), $\tau = 1000$ (purple down triangles).

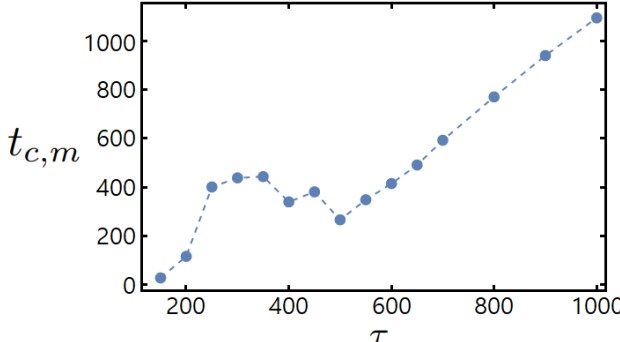

Figure 6: Critical time $t_{c,m}$ as a function of $\tau$, for $r = 320$ and $E_0 = 0.14$.

durations $\tau$. We note that the critical time increases with $r$, as already highlighted in Fig. 4, but it is not monotonous with $\tau$. This may look counterintuitive, since one expects that a longer laser pulse at fixed $E_0$ transfers more energy from subsystem 1 to subsystem 2. However, under the effect of the pulse perturbation, the energy of subsystem 1 exhibits an oscillatory behaviour during its early time evolution. It follows that, if the perturbation lasts a time too short for dissipation to fully set in, its final effect critically depends whether, at the pulse end, the energy has reached a maximum or a minimum of its evolution. On the contrary, if the pulse duration is longer than the typical timescale of dissipation, such memory effect is lost. To stress this behaviour, in Fig. 6 we plot $t_{c,m}$ as a function of $\tau$ at fixed $E_0 = 0.14$ and $r = 320$. We observe that for $\tau < 500$ the critical time is not monotonous, while it becomes so only for larger $\tau$, where it grows linearly with the pulse duration.

To complete our analysis of the 'critical time' dependence upon the bath and pulse parameters, in Fig. 7 we plot $t_{c,m}$ as function of $E_0$, at fixed $r = 640$ and for three different values of $\tau$. In conclusion, when the pulse duration is long enough to make dissipation active well before the pulse end, the time $t_{c,m}$ during which the system is trapped into a non-thermal symmetry broken state increases monotonously with $r$, $\tau$ and $E_0$.

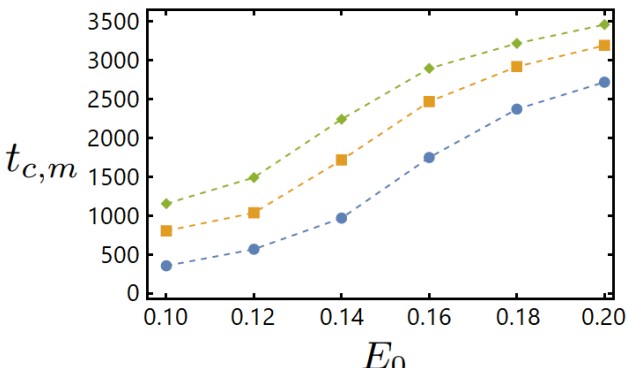

Figure 7: Critical time $t_{c,m}$ as a function of $E_0$, for $r = 640$ and: $\tau = 750$ (blue curve), $\tau = 1000$ (orange curve), $\tau = 1200$ (green curve).

## 3.2   Time evolution at constant pulse 'fluence'

Until now, we have compared results obtained for perturbations having the same amplitude $E_0$ or the same duration $\tau$. In what follows, we study the system response upon increasing the laser pulse duration $\tau$ while properly reducing its peak amplitude so as to maintain constant the total supplied energy, defined as [24]

$$F = \int_0^\infty |E(t)|^2 \, dt, \tag{28}$$

which can be regarded as the pulse 'fluence'. In Figs. 8 and 9 we show the time evolution of the order parameter, $m_{x,1}$, for increasing $\tau$ at constant $F = 15.53$, thus decreasing $E_0$ correspondingly.

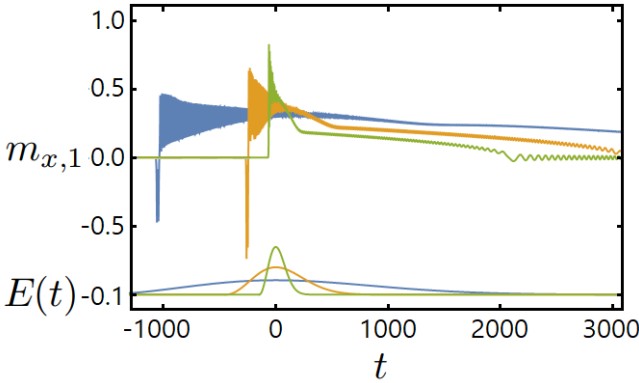

Figure 8: Time evolution of the order parameter, $m_{x,1}$, at $r = 640$ and different values of $\tau$ and $E_0$ such that $F$ in Eq. (28) is kept constant at the value 15.53. In particular, we use $\tau = 5000$ and $E_0 = 0.105$ (blue curve), $\tau = 1000$ and $E_0 = 0.2$ (orange curve), $\tau = 250$ and $E_0 = 0.35$ (green curve). The curves have been shifted so that $t = 0$ corresponds to the peak amplitude of the pulses, shown in the lower part of the plot.

We observe that, while for short times $m_{x,1}$ peaks more in presence of a spiked pulse rather than a longer but flatter one, for long times the latter is much more efficient to make a finite $m_{x,1}$ survive longer. Fig. 10 shows that the critical time, $t_{c,m}$, indeed grows substantially with $\tau$ at fixed $F$. Essentially, for large $\tau$, the transient non-thermal symmetry-broken phase becomes a quasi-steady state kept alive by the dissipative bath.

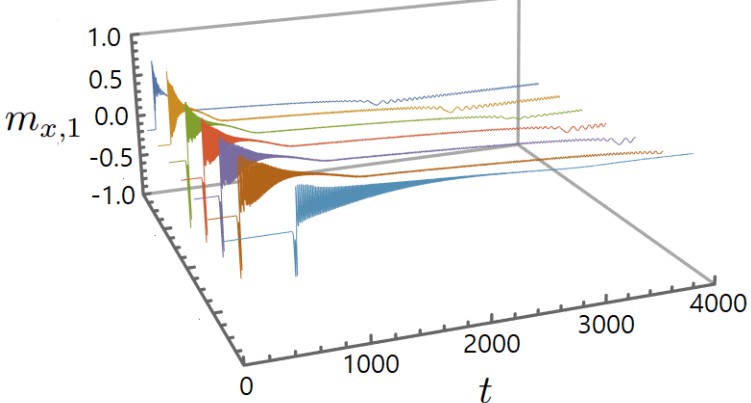

Figure 9: Same as in Fig.8 without the time shift, and for $\tau = 250$ (blue curve), $\tau = 500$ (orange curve), $\tau = 750$ (green curve), $\tau = 1000$ (red curve), $\tau = 1250$ (purple curve), $\tau = 1500$ (brown curve), $\tau = 5000$ (light blue curve).

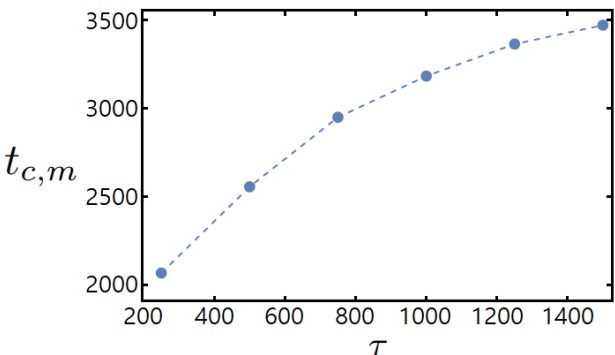

Figure 10: Critical time $t_{c,m}$ as a function of $\tau$. The amplitude $E_0(\tau)$ is chosen so as to maintain constant the 'fluence' in Eq. (28).

## 4   Conclusions

In this paper we have investigated the transient cooling mechanism brought on by a pulse perturbation in the two coupled infinite-range quantum Ising models of Ref. [21], but now in presence of dissipation. We have shown that the cooling of the low-energy degrees of freedom at the expense of the high-energy ones, observed in absence of dissipation, is not spoiled in its presence, especially when excitations dissipate faster the higher their energy. On the contrary, dissipation enhances the cooling effect of the perturbation, stabilising a non-thermal quasi-steady state that lasts for long after the pulse end. Specifically, we have found that increasing the pulse duration keeping the 'fluence', $F$ of Eq. (28), constant, makes such quasi-steady state survive longer and longer, not in disagreement with recent experiments in $K_3C_{60}$ [14].
Such dissipative cooling effect resembles much the nuclear Overhauser effect [25–27], which has been also realised by optical pumping [28], and, to some extent, laser cooling in optomechanical systems, where mechanical oscillators can be cooled by coupling them to a microwave cavity [29–31]. However, in our case the role of entropy source and sink are played by the low- and high-energy internal degree of freedom of the system, without the need of an optical or microwave cavity. As a consequence, in our model the transient cooling does not suffer from the stringent limitations given, e.g., by the microwave cavity decay

rate, which strongly affect optomechanical cooling, and make challenging its experimental realisation [32].

**Funding information**   A. N. was financially supported by POR Calabria FESR-FSE 2014/2020 - Linea B) Azione 10.5.12, grant no. A.5.1. M. F. acknowledges financial support from the European Research Council (ERC) under the European Union's Horizon 2020 research and innovation programme, Grant agreement No. 692670 "FIRSTORM".

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
