# Peer review of "Dissipative cooling induced by pulse perturbations"

_SciPost Physics_

## Round 1 · Referee Report · Anonymous · 2021-7-2

Report
Nava and Fabrizio consider the dynamics of two coupled transverse
field quantum Ising models subject to pulsed perturbation. The
authors consider the case in which all spins belonging to a given
Ising model interact via an all-to-all interaction. This property
makes the system amenable for an analytical analysis. The idea of
the paper builds largely on the work [PRL 120, 220601 (2018)] by
one of the authors. In comparison to this previous work the authors
consider now also dissipative effects which are modelled by a
Lindblad master equation.
I do not think that the paper is sufficiently innovative and novel
to merit publication in SciPost. Moreover, I find that the paper is
not very easy to access and written in a contrived way.
Specific comments:
1) I wonder why the authors are introducing the hard core bosons in
Eq. (15). They do not seem to be used later on, and the discussion
concerning the structure of the spectrum [Eqs. (14)] could also be
performed without them. Is there any profound reason for the
introduction of hard core bosons?
2) I do not quite understand why Eq. (18) is mimicking a laser
pulse. In my opinion a laser acts at the level of a single particle
(due to the dipolar selection rule). Therefore the corresponding
operator should be a sum of single body operators. Where does the
\sigma_x \sigma_x - term come from? This rather resembles a (dipolar)
interaction.
3) The notation of Eq. (23) is confusing when compared with Eq.
(8). In one expression the argument is time while in the other one
it is temperature.
4) I find the motivation of the dissipative terms not very
convincing. Is it justified to assume that the rates are explicitly
time-dependent, i.e. isn't there some implicit separation of time
scales that is assumed to hold?
5) In the conclusions you write "On the contrary, dissipation
enhances the cooling effect of the perturbation, stabilising a
non-thermal quasi-steady state that lasts for long after the pulse
end." Are the authors sure that the dissipation is not just
constructed in a way that this is the case. There are a number of
assumptions, e.g. concerning the ratio of the rates, which are to
some extend arbitrary. Notes, that the relative energy difference
is only one of the quantities that is entering the transition
rates. There is also a kinetic part which determines whether a
transition can take place (states with large energy difference may
be connected by complicated transition paths). Especially for
correlated systems this seems to be a relevant aspect. In this
sense I find the connection to photo-induced superconductivity in
K3C60 also a bit far-fetched.

---

## Round 1 · Referee Report · Anonymous · 2021-7-2

Strengths
1. The manuscript convincingly presents an interesting result, namely the appearance of transient low-entropy states of a driven many-body system under the inclusion of thermalization dynamics.
2. The mechanism studied in this work could provide an understanding of laser-induced superconductivity.
3. The manuscript is very accessible and well-written.
Weaknesses
It is a bit hard to judge whether the effects observed by the authors are generic properties of driven many-body system or just limited to the toy model being studied.
Report
The manuscript by Nava and Fabrizio studies a driven many-body system under the influence of thermalization dynamics in Lindblad form. Similar to the previous work in [20], the authors suggest that long-range Ising models could serve as a toy model to study laser-induced cooling of many-body systems, which could be relevant to explain laser-induced superconductivity. Here, the authors also include thermalization dynamics to make the model more realistic. Combining the study of driven many-body systems with open system dynamics opens an interesting direction into the subject and in principle fulfills the acceptance criteria of SciPost Physics. However, there are some questions that the authors should address before publication:
1. The idea to use driven-dissipative dynamics to cool strongly interacting many-body systems has also been investigated recently in the context of reservoir engineering techniques (e.g., [New J. Phys. 15, 073027; Science Adv. 6, eaaw9268; Phys. Rev. Research 2, 023214]). The authors should discuss how their approach is related the these works.
2. What is the total area of the pulse in the parameterization chosen by the authors? Is it possible for the energy to flow back to the system 1 or do the authors consider the equivalent of a \pi pulse between the two systems?
3. Is the critical time somehow related to the critical properties of the underlying phase transition of the Ising model?
Requested changes
see Report

---

## Round 1 · Referee Report · Anonymous · 2021-7-5

Strengths
1- interesting original ideas
Weaknesses
1- colloquial language and short explanations
2- little discussion of implications of results
Report
The manuscript describes an interesting aspect of laser-cooling of many-body systems.
Even though the work is a continuation of existing work, I do think that there is enough original material that justifies publication.
I am not sure how realistic the considered model of two coupled Ising Hamiltonians is; maybe the authors can comment on that in a revised manuscript.
I feel that the discussion in the manuscript is a bit brief. The authors describe their analysis and they show results in figures, but the discussion of implications of their work is very brief. Also, the language is often very colloquial and many expressions are ambiguous and open to interpretation; e.g.: 'firing a laser pulse', 'soak up', 'deprived of entropy', 'does not spoil Eq.(3)'.
I feel that a carefully revised manuscript with extended discussion (in particular of the data shown in the Figs) and more precise language could be a valuable contribution.
I would be inclined to give a positive recommendation for a properly revised manuscript.

---

## Editorial Decision

unknown